# Multiclass classification of Autism Spectrum Disorder, attention deficit hyperactivity disorder, and typically developed individuals using fMRI functional connectivity analysis

Caroline L. Alves[1]*, Tiago Martinelli[2], Loriz Francisco Sallum[2], Francisco Aparecido Rodrigues[2], Thaise G. L. de O. Toutain[3], Joel Augusto Moura Porto[4,5], Christiane Thielemann[6], Patrícia Maria de Carvalho Aguiar[7,8], Michael Moeckel[1]

1 Laboratory for Hybrid Modeling, Aschaffenburg University of Applied Sciences, Aschaffenburg, Bayern, Germany, 2 Institute of Mathematical and Computer Sciences, University of São Paulo, São Paulo, São Paulo, Brazil, 3 Health Sciences Institute, Federal University of Bahia, Salvador, Bahia, Brazil, 4 Institute of Physics of São Carlos (IFSC), University of São Paulo (USP), São Carlos, São Paulo, Brazil, 5 Institute of Biological Information Processing, Heinrich Heine University Düsseldorf, Düsseldorf, North Rhine–Westphalia Land, Germany, 6 BioMEMS Lab, Aschaffenburg University of Applied Sciences, Aschaffenburg, Bayern, Germany, 7 Hospital Israelita Albert Einstein, São Paulo, São Paulo, Brazil, 8 Department of Neurology and Neurosurgery, Federal University of São Paulo, São Paulo, São Paulo, Brazil

* CarolineLourencoAlves@th-ab.de

**Data Availability Statement:** All relevant data are within the article and its Supporting information

## Abstract

Neurodevelopmental conditions, such as Autism Spectrum Disorder (ASD) and Attention Deficit Hyperactivity Disorder (ADHD), present unique challenges due to overlapping symptoms, making an accurate diagnosis and targeted intervention difficult. Our study employs advanced machine learning techniques to analyze functional magnetic resonance imaging (fMRI) data from individuals with ASD, ADHD, and typically developed (TD) controls, totaling 120 subjects in the study. Leveraging multiclass classification (ML) algorithms, we achieve superior accuracy in distinguishing between ASD, ADHD, and TD groups, surpassing existing benchmarks with an area under the ROC curve near 98%. Our analysis reveals distinct neural signatures associated with ASD and ADHD: individuals with ADHD exhibit altered connectivity patterns of regions involved in attention and impulse control, whereas those with ASD show disruptions in brain regions critical for social and cognitive functions. The observed connectivity patterns, on which the ML classification rests, agree with established diagnostic approaches based on clinical symptoms. Furthermore, complex network analyses highlight differences in brain network integration and segregation among the three groups. Our findings pave the way for refined, ML-enhanced diagnostics in accordance with established practices, offering a promising avenue for developing trustworthy clinical decision-support systems.

files. Neuroimaging data were obtained from two large, publicly available datasets: ADHD-200 (http://fcon_1000.projects.nitrc.org/indi/adhd200/) and ABIDE (Autism Brain Imaging Data Exchange; http://fcon_1000.projects.nitrc.org/indi/abide/).

**Funding:** ZeWiS is funded through public research projects by the Bavarian State Ministry for Sciences and the Arts. The funder provided support in the form of salaries for authors but did not have any additional role in the study design, data collection and analysis, decision to publish, or manuscript preparation. The specific roles of these authors are articulated in the 'author contributions' section.

**Competing interests:** EpilabKI is funded through the Bavarian stated Ministery for Sciences and the Arts research. This does not alter our adherence to PLOS ONE policies on sharing data and materials.

# 1 Introduction

## 1.1 Clinical background

Neurodevelopmental disorders encompass a spectrum of conditions that manifest early in life and have diverse impacts on brain development and function, often presenting with genetic and clinical heterogeneity [1]. These disorders profoundly affect neurological functioning, including cognition, communication, behavior, motor skills, and social interaction [2–4].

Two prominent examples of neurodevelopmental disorders are Autism Spectrum Disorder (ASD) and Attention Deficit Hyperactivity Disorder (ADHD). ASD is characterized by challenges in social interaction, communication, repetitive behaviors, and sensory sensitivities [5]. Globally, ASD affects approximately 1 in 36 children and is more prevalent in males than females [6]. ASD is a spectrum disorder, displaying a wide range of symptom severity and presentation, making diagnosis a challenging task [7–9]. Furthermore, ASD is marked by significant heterogeneity, with no discernible patterns consistently emerging among affected individuals [10].

ADHD, another prevalent neurodevelopmental disorder, is defined by inattention, hyperactivity, and impulsivity symptoms, which can substantially impact daily functioning [11]. It affects 5–8% of children, with a higher prevalence among boys [12]. Despite extensive research, the prevalence of ADHD remains elusive [13], and diagnosis primarily relies on the assessment of behavioral symptoms [11].

While ASD and ADHD are traditionally classified as distinct neurodevelopmental disorders, they exhibit a significant degree of symptom overlap [14]. This shared symptomatology often complicates the accurate diagnosis and treatment planning for affected individuals. Furthermore, it is noteworthy that ADHD frequently co-occurs with ASD, making it one of the most prevalent comorbidities among individuals with ASD [15]. This comorbidity adds another layer of complexity to the neurodevelopmental profile of affected individuals, contributing to the challenges in diagnosis and care. Consequently, these circumstances often lead to cases of misdiagnosis and underdiagnosis.

## 1.2 Previous ML approaches

Given the inherent complexity of diagnosing ASD and ADHD, many studies are using machine learning methods to improve the diagnosis [16]. In the study by [17], machine learning models were trained and tested on an imbalanced dataset from research records based on the Social Responsiveness Scale (SRS). The SRS is a parent-administered questionnaire frequently employed for measuring autism traits. Notably, the superior performance algorithms were Support Vector Machines (SVM), Ridge Regression, Least Absolute Shrinkage and Selection Operator (LASSO), and Linear Discriminant Analysis, achieving accuracies ranging from 0.962 to 0.965. These results underscore the effectiveness of machine learning techniques in accurately distinguishing individuals with ASD from those with ADHD. Moreover, in [18] it was incorporated a crowdsourced dataset comprising responses to 15 SRS-derived questions. In this subsequent analysis, Linear Discriminant Analysis (LDA) and Elastic Net (ENet) emerged as the top-performing methods, both achieving an Area Under the ROC Curve (AUC) of 0.89 when tested on survey data containing the 15 questions.

Machine methods have also been applied to neuroimaging data to develop more precise and reliable approaches for characterizing and predicting ASD and ADHD in a binary way to distinguish from TD [11, 13, 19–24]. Most studies in the literature are primarily focused on distinguishing individuals with a specific condition from typically developing (TD) individuals, resulting in a binary classification problem. However, a more desirable scenario would

**Table 1. Studies with ML classification algorithms for distinguishing ASD and ADHD groups described in subsection 1.2.**

| Studies | Dataset type | ML algorithm | Classification type | Correlation metric | AUC | Accuracy | Recall | Precision |
|---------|-------------|--------------|---------------------|--------------------|-----|----------|--------|-----------|
| [17] | research records survey | SVM, Ridge Regression, LASSO, and LDA | Binary (ASD and ADHD) | - | - | 0.962–0.965 | - | - |
| [18] | research records survey | LDA and ENet | Binary (ASD and ADHD) | - | 0.890 | - | - | - |
| [25] | research records survey | SVM | Binary (ASD and ADHD) | - | 0.910 | - | 0.680 | - |
| [26] | MRI | GPC | Multiclass (ASD, ADHD and TD) | - | - | 0.682 | - | - |
| [30] | fMRI | SVM | Multiclass (ASD, ADHD and TD) | Pearson correlation | - | 0.66 | 0.82 | 0.59 |

involve the differentiation among multiple neurological conditions characterized by overlapping symptoms or co-occurrence, as seen in the case of ASD and ADHD. Preliminary efforts have been made in this direction, in which various machine learning techniques have been employed [17, 18, 25]. For instance, one of the pioneering studies to differentiate adolescents with ADHD from those with ASD or TD was published in 2013 [26]. The authors considered structural magnetic resonance imaging (MRI) data and a 3-class Gaussian Process Classification (GPC) approach to classify ADHD, ASD, and TD groups simultaneously [26]. Their model achieved a balanced accuracy of 0.682, with sensitivities of 0.759, 0.655, and 0.632 for the ADHD, TD, and ASD groups, respectively. The positive predictive values for the respective groups were also 0.629, 0.731, and 0.75.

In [27], centrality abnormalities in cortical and subcortical regions were discovered, some of them shared between ADHD and ASD. They observed increased centrality in the right striatum/pallidum for ADHD and bilateral temporolimbic areas for ASD. In [28], it was conducted a comparative analysis of the network topology patterns among ASD, ADHD, and neurotypical (NT) groups. They found substantial overlap at the global level of community structure among all groups. However, the overlap between the two clinical conditions was less than that between each condition and the control group.

Additionally, the ASD and ADHD groups exhibited a more pronounced reduction in correlation strength with increased distance compared to the NT group. Notably, the ADHD group displayed reduced wiring costs, thinner cortical regions, and lower hub degrees than the ASD group. Significant findings emerged in the study [29] regarding oscillatory patterns in children with ASD and ADHD during task conditions. Children with ASD exhibited significant hypoconnectivity in large-scale networks during these tasks, while those with ADHD showed hyperconnectivity in large-scale networks under similar conditions. In a recent study [30], an SVM algorithm with l2-regularization emerged as the top-performing method, achieving an accuracy of 0.66, an f1-score of 0.68, a precision of 0.59, and a recall of 0.82. Notably, their findings unveiled a substantial convergence in functional brain connectivity patterns between ADHD and ASD, particularly within the right ventral attention network, the salience network, and the default mode network (DMN) as observed in resting-state fMRI data.

Table 1 concisely overviews the primary research using machine learning classification methods and the ASD and ADHD groups outlined in this subsection.

### 1.3 Research gap

Previous ML models have yet to be constructed using explainable AI approaches. This limits their interpretability and, hence, the trustworthiness of their model predictions. In particular,

a clear connection between established clinical symptoms and model properties has yet to be drawn, raising doubts among medical professionals and hindering the use of ML modeling in actual clinical diagnosis [31].

Upon reviewing the landscape of studies within the domain of ASD and ADHD classification, a notable trend emerges, as illustrated in the summarized literature (Table 1): a predominant focus on binary comparisons between ASD and ADHD, often reliant on survey-based symptomatic datasets, which not alllows clear separation in the cases that have overlapping of symptoms. This emphasis on a binary framework potentially oversimplifies the nuanced complexities inherent in these neurodevelopmental disorders. Moreover, the reliance on survey data in many studies raises concerns regarding introducing biases [32], particularly when compared to more direct neuroimaging modalities such as EEG and fMRI. Our previous research [33, 34] highlights the critical role of correlation metrics in constructing connectivity matrices to effectively capture brain changes associated with these conditions. While studies such as [30] use fMRI and a multiclass approach, they predominantly utilize linear measures like Pearson correlation; the robustness of alternative metrics such as normalized transfer entropy still needs to be explored. In our prior works [34, 35], we demonstrated the robustness of normalized transfer entropy in distinguishing neurodevelopmental disorders from TD individuals, underscoring the need to explore these metrics further. Furthermore, the prevailing focus on ML methodologies overlooks the exploration of complex network topology changes and needs more medical interpretation. There is a clear need for a more holistic approach that integrates diverse methodologies and prioritizes nuanced understanding over simplistic binary classifications. Such an approach holds the potential to advance our comprehension of the underlying mechanisms of ASD and ADHD and inform more effective interventions and treatments.

### 1.4 Objective and hypothesis

Our study aims to bridge the existing research gap by advancing beyond simplistic binary comparisons in the classification of ASD, ADHD, and TD individuals using fMRI datasets. Building upon prior work that underscores the limitations of binary frameworks and the potential bias introduced by survey-based datasets, our hypothesis posits that distinct brain activity patterns underlie these neurodevelopmental disorders and can facilitate their reliable separation. We seek to investigate whether these patterns align with existing clinical knowledge of ASD and ADHD, thereby providing deeper insights into these conditions' underlying mechanisms and proving our methodology's trustworthiness.

Additionally, we aim to explore the utility of complex network measures in achieving a clean separation of the groups, surpassing the conventional focus on machine learning methodologies, through the integration of diverse methodologies alongside advanced analytical techniques such as normalized transfer entropy, our study endeavors to enhance prediction accuracy while ensuring an explainable and trustworthy machine learning approach.

## 2 Materials and methods

### 2.1 Innovations in the methodology

The current paper endeavors to investigate the feasibility of automatically detecting brain changes associated with ASD and ADHD while simultaneously providing a biological rationale for these observations. We leverage the blood oxygenation level-dependent (BOLD) time series data to achieve this objective and develop a classification method to distinguish individuals with ASD, ADHD, and TD profiles. Following dataset preprocessing (described in subsection 2.2), we employed two levels of data abstraction: (A) the calculation of correlations, determined by the normalized transfer entropy between specific fMRI regions of interest (described

in subsection 2.3), and (B) the extraction of complex network measures from the correlations (A) (described in subsection 2.4). It is important to note that while our methodology bears similarity to our previous work [33–37], where binary classification was primarily explored, the present study aims to establish a multiclass classifier. Moreover, departing from existing literature, which often focuses on utilizing only one of these abstraction levels, our study pioneers the simultaneous use of both levels in a multiclass context, marking a novel contribution to the field.

To enhance the interpretability of our machine learning results, we incorporated cutting-edge techniques that have emerged in recent years. One such technique is the application of SHapley Additive ExPlanations (SHAP) values [38]. These values help identify the most critical features within our model, shedding light on essential brain areas and connections between regions. Notably, SHAP values have demonstrated superior performance compared to prior research efforts [33, 36, 39] in pinpointing significant brain areas and connectivity patterns and have been an integral part of our previous work.

Differently from [33–37], we added three measures to analyze the segregation and integration concepts: Effective Information, determinism, and degeneracy coefficients. These measures provide a comprehensive understanding of the dynamics of brain networks in individuals with ASD, ADHD, and TD profiles. Furthermore, to the best of our knowledge, this is the first study that employed the SHAP values methodology for a multiclass classification of ASD and ADHD, enhancing the interpretability and robustness of our classification results.

The Python code with the methodology used in this work and described in this section is available at: https://github.com/Carol180619/Multiclass-ADHD-ASD.git.

## 2.2 Data and data preprocessing

The ADHD dataset used in this study was sourced from the Neuro Bureau ADHD-200 Preprocessed repository, as detailed in [40]. During a 6-minute resting-state fMRI scan, participants received instructions to relax, maintain closed eyes, avoid falling asleep, and refrain from engaging in specific thoughts. The resting-state fMRI data captures spontaneous fluctuations in BOLD signals, widely acknowledged as indicative of underlying brain activity. In this investigation, we utilized the ADHD-200 dataset via the Nilearn package. Nilearn is a Python package [41] tailored explicitly for analyzing neuroimaging data. Our selection of Nilearn seamlessly integrated with our existing workflow, as we had already incorporated Nilearn into our analysis pipeline. Nilearn provides a comprehensive set of tools for preprocessing, feature extraction, and statistical analysis of neuroimaging data. It was used in numerous studies [42–44], making it a suitable and convenient choice for our research. Within the Nilearn package, the dataset comprises 40 child and adolescent participants aged 7 to 27, equally divided between individuals diagnosed with ADHD and healthy control subjects.

As in our previous work [34], we considered the preprocessed version of the Autism Brain Imaging Data Exchange (ABIDE) (Avaiable in https://fcon_1000.projects.nitrc.org/indi/abide/) which consists of 1112 datasets comprised of 539 ASD and 573 TD with 300s BOLD time series (7–64 years, median 14.7 years across groups). Further, it is also available for use in Nilearn's Python package. We used the same amount of 40 subjects from the ADHD dataset. For TD groups, we randomly selected 20 subjects for the ABIDE dataset and 20 for the Neuro Bureau ADHD-200 Preprocessed repository.

In our study, rather than utilizing the entire BOLD time series obtained from each voxel in brain images, we focused on specific Brain Regions of Interest (ROIs). These ROIs are defined based on a brain atlas; only the BOLD time series from these ROIs are considered. The choice

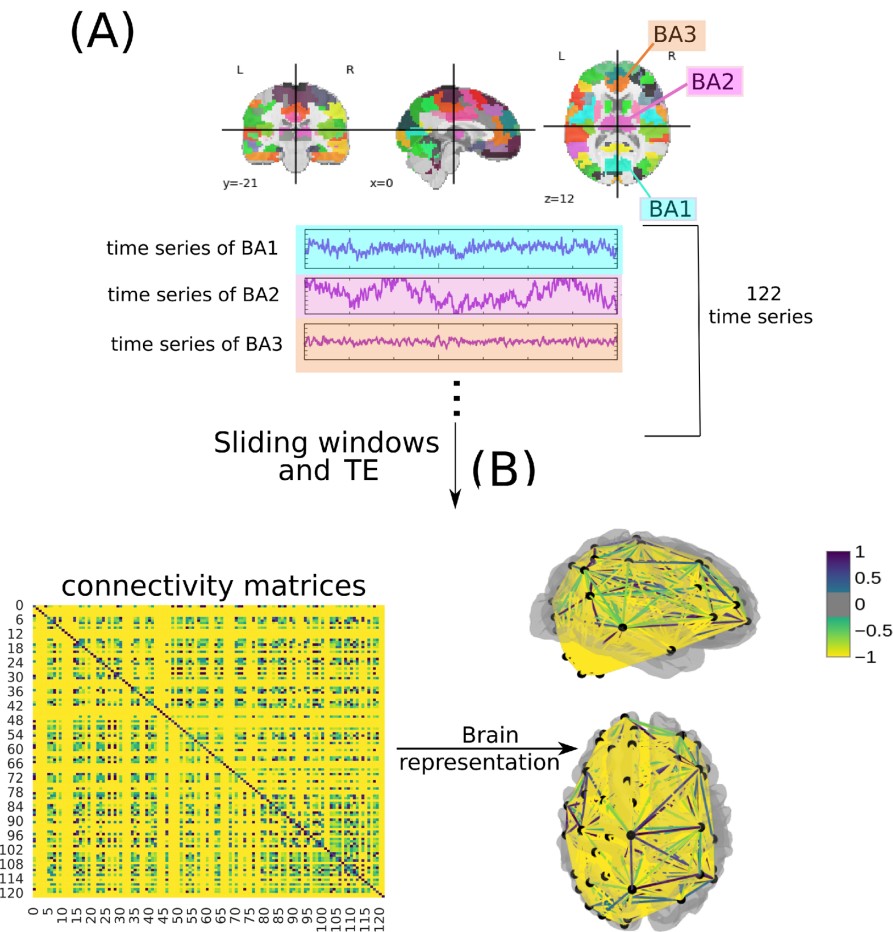

**Fig 1. Methodology to obtain the connectivity matrices based on** [34]**.** In (A), a time series of 122 ROI is extracted from fMRI data using the BASC BOLD atlas (highlighted in blue, purple, and orange). A sliding window was performed as a data augmentation. Then, they are correlated (B) to form the connectivity matrices, where each row and column corresponds to one of the Brodmann areas for a patient with ASD, TD, and ADHD (the figure illustrates an example of a connectivity matrix with a normalized TE of a subject presenting ADHD). The same highlighted matrices represent the brain in a three-dimensional (in a top and left perspective) schematic.

of Brain Atlas is crucial, and in our work, we employed the Bootstrap Analysis of Stable Clusters (BASC) atlas, selected for its exceptional performance in discriminating Autism Spectrum Disorder (ASD) patients using deep learning models, as reported in [45]. The BASC atlas, introduced initially in [46], is derived from group-level brain parcellation through the BASC method, an algorithm utilizing k-means clustering to identify brain networks with coherent activity in resting-state fMRI, as described in [47]. BASC map with a cluster number of 122 ROIs is used here (see Fig 1A). From our previous work [34], a manual use of Yale BioImage Suite Package web application (Avaiable in https://bioimagesuiteweb.github.io/webapp/mni2tal.html) labeled the coordinates of each ROI for the identification of their names (see Fig 1A).

After extracting the BOLD time series, a sliding time window of 20 seconds was employed for data augmentation. This duration was selected based on our previous study [34], where it demonstrated optimal performance for the ASD dataset. Additionally, to ensure comparability between the ASD and ADHD datasets, the same window size was

utilized for the ADHD dataset. By employing consistent time windows across both datasets, we aimed to mitigate potential biases arising from variations in data acquisition protocols between different sites and enhance the robustness of our analyses. Through the data augmentation process, 600 matrices were randomly selected, ensuring an equal representation of each class.

Once the time series for each of the 122 regions had been extracted for each time window, they were correlated according to normalized transfer entropy (TE)(see Fig 1B). The TE is described in [48] and given by the Eq 1. The normalized TE metric was selected as our study's primary analytical tool, building upon prior research findings [34, 35]. TE has demonstrated its efficacy in capturing changes in brain dynamics associated with neurodevelopmental disorders. This choice was motivated by several advantages it offers. Firstly, normalized TE is adept at capturing directional dependencies [49], providing a nuanced understanding of information flow within neural systems [50, 51]. Unlike linear measures, such as Spearman and Pearson correlation, TE is non-linear and model-free, making it particularly suitable for analyzing complex, non-stationary, and non-Gaussian datasets [52–54]. Moreover, TE stands apart from traditional correlation measures like the Pearson correlation. While the latter is confined to assessing linear or monotonic relationships between variables, regardless of their direction of influence, TE excels in identifying and quantifying the directional dependencies between time series [55], thereby facilitating a more comprehensive assessment of neural interactions. Noteworthy, as in our previous work [34, 35], for computing TE, a Min-max normalization and then a thresholding process was performed, with a value of 0.5, since this measures deal best with binary values.

$$TE_{Y \to X}(\tau) = \sum_{i=1}^{N} P(x_{t+\tau}^i, x_t^i, y_t^i) \log \frac{P(x_{t+\tau}^i | x_t^i, y_t^i)}{P(x_{t+\tau}^i | x_t^i)} \tag{1}$$

where:

- $TE_{Y \to X}(\tau)$ denotes the transfer entropy from $Y$ to $X$ at time lag $\tau$.

- $x_t^i$ and $y_t^i$ represent samples from time series $X$ and $Y$ respectively.

- $N$ is the total number of samples.

- $P(x_{t+\tau}^i, x_t^i, y_t^i)$ is the joint probability distribution of $x_{t+\tau}^i$, $x_t^i$, and $y_t^i$.

- $P(x_{t+\tau}^i | x_t^i, y_t^i)$ is the conditional probability distribution of $x_{t+\tau}^i$ given $x_t^i$ and $y_t^i$.

- $P(x_{t+\tau}^i | x_t^i)$ is the conditional probability distribution of $x_{t+\tau}^i$ given $x_t^i$.

In addition to the aforementioned preprocessing steps, we employed the NeuroImaging Analysis Kit (NIAK) [56] to further standardize and enhance the quality of our neuroimaging data for both datasets. NIAK offers a comprehensive set of tools for preprocessing fMRI data, including motion correction, slice timing correction, spatial normalization, and nuisance signal regression [40, 57]. These preprocessing procedures are crucial for mitigating potential confounding factors introduced by differences in data acquisition protocols across multiple sites. By implementing the same preprocessing pipeline in both datasets, we aimed to minimize site-related variations and ensure the consistency and reliability of our data across different acquisition sites. This standardized approach facilitated the integration of neuroimaging data from disparate sources, enhancing the validity and generalizability of our findings.

## 2.3 Connectivity matrices

Similar to our prior research endeavors [33–37], we leveraged Machine Learning (ML) algorithms to analyze fMRI data at different levels of abstraction the connectivity matrix (A) and the attribute matrix (B), which comprises complex network metrics derived from (A). To conduct our analysis, we employed a diverse set of ML classifiers, including the Support Vector Machine (SVM) [58], Naive Bayes (NB) [59], Multilayer Perceptron (MLP) [60], a fine-tuned Convolutional Neural Network (CNN), and Long Short-Term Memory neural networks (LSTM) [61].

Subsequently, the SHAP values method was employed for the biological interpretation, as it provides a comprehensive explanation of the predictive contribution of each feature. To ensure robustness and unbiased assessment of the machine learning models, we adopted a consistent evaluation approach: 10-fold stratified cross-validation with shuffling. This cross-validation technique partitions the dataset into ten equal stratified folds, ensuring that each fold, denoted as *k*, maintains a balanced distribution of samples from each class. This approach guarantees that all classes are equally represented across the folds, thus enhancing the reliability of our model evaluations. The algorithm then trains on nine of these folds and validates on the remaining fold, repeating this process ten times, each serving as the validation set once. We used k = 10, which is a common value for this method [62–66]. However, to show the stability of the model's performance, we also tested different values of k. Moreover, the shuffle strategy ensures the data is randomized before being split into folds to prevent ordering effects; this randomization helps prevent any potential bias stemming from data ordering, ensuring the robustness of the model training process.

Furthermore, as a crucial step in our experimental methodology, we performed an initial partition of 30% of the original dataset (comprising a total of 600 matrices) for final testing, constituting an exclusive reserve of 180 matrices. This separation is done before the model's training using a 10-fold cross-validation. Adopting such a practice is commonplace in machine learning, and it serves the dual purpose of assessing model performance while mitigating overfitting and ensuring its ability to generalize to new, unseen data [67, 68].

This procedure was applied for model selection and hyperparameter optimization. It was also considered the grid search method, commonly used in the literature [69–73], used for all ML algorithms except the CNN and LSMT models. In the deep learning models, we used random search optimization because it offers a more computationally efficient alternative for hyperparameter tuning compared to grid search, which is particularly advantageous, given the high computational demands of deep learning tasks. The hyper-parameter optimization values for each classifier model can be seen in more detail in [33–36]. In Tables 2 and 3, we present the architectural details and hyperparameter configurations for CNN and LSTM models, respectively. Notably, dropout regularization, as indicated in Tables 2 and 3, is a widely-used technique in neural network training to combat overfitting [74]. Dropout operates by randomly deactivating neurons during training, compelling the network to learn more resilient and generalizable features [75]. Empirically, dropout has effectively enhanced the generalization capabilities of deep learning architectures and medical data [76, 77].

Additionally, in our training set, we applied a process known as standardization. Standardization in machine learning typically involves rescaling features to have a mean of zero and a standard deviation of one [78]. This step is pivotal as it transforms the data, facilitating more straightforward comparisons and analyses [79]. This practice is essential because it ensures that all model attributes are equally important and share a consistent scale [80]. Moreover, standardization safeguards against the undue influence of outliers and features with substantial variability on the model's performance [81].

**Table 2. Hyperparameters and layer configurations for the CNN model.**

| Type of Layer | Tuning hyperparameter | Value |
|---|---|---|
| Convolutional | — | — |
| Convolutional | dropout | [0.00, 0.05, 0.10, 0.15, 0.20, 0.25, 0.30, 0.35, 0.40, 0.45, 0.50] |
| Convolutional | — | — |
| Convolutional | number of filters | [32, 64] |
| Max Pooling | dropout | [0.00, 0.50, 0.10, 0.15, 0.20] |
| Flatten | — | — |
| Dense | - units | [32, 64, 96. . . .512] |
| | -activation | [relu, tanh, sigmoid] |
| Dropout | rate | [0.00, 0.50, 0.10, 0.15, 0.20] |
| Adam optimization compile | learning rate | $min - value = 1e^{-4}$ |
| | | $max - value = 1e^{-2}$ |
| | | sampling = LOG |

First and foremost, we considered the widely-used accuracy metric, which provides an overall assessment of our classification model's correctness [82–86]. We expanded our evaluation to incorporate additional standard metrics such as precision and recall [87–90]. Precision, also known as a positive predictive value, measures our model's ability to classify instances belonging to a specific class correctly. In our case, precision helps gauge our model's accuracy in identifying the TD group. On the other hand, recall, also known as sensitivity, assesses how effectively our classifier predicts positive examples, which now encompass ASD and ADHD individuals. To visualize the performance of our classification model, we continued to utilize the Receiver Operating Characteristic (ROC) curve, a standard method for illustrating the trade-off between true and false positive rates. The Area Under the ROC Curve (AUC) remained a key evaluation metric, with values ranging from 0 to 1. An AUC of 1 signifies a flawless classification, while 0.5 suggests a random choice where the classifier cannot distinguish between classes effectively [72, 82, 91, 92].

In this three-class classification context, we calculated the micro-average AUC independently for each class (TD, ASD, or ADHD) to provide insights into individual class performance. This micro-average computation treats each class equally, allowing us to assess how well our model performs for each group. Furthermore, we introduced the concept of macro average in our evaluation, which considers the classes individually and aggregates their contributions before calculating the average. Unlike the micro average, the macro average does not

**Table 3. Hyperparameters and layer configurations for the LSTM model.**

| Type of Layer | Tuning hyperparameter | Value |
|---|---|---|
| LSTM | — | — |
| LSTM | dropout | [0.00, 0.05, 0.10, 0.15, 0.20, 0.25, 0.30, 0.35, 0.40, 0.45, 0.50] |
| LSTM | — | — |
| LSTM | units | [70, 60, 50, 40] |
| Dense | - units | 3 |
| | -activation | softmax |
| Adam optimization compile | learning rate | $min - value = 1e^{-10}$ |
| | | $max - value = 1e^{-1}$ |
| | | sampling = LOG |

treat all classes equally, providing a different perspective on the overall performance of our classification model. Subsequently, the SHAP values method was used for the biological interpretation, as it explains the predictive power of each attribute. The SHAP values method was subsequently employed for the biological interpretation, as it explains the predictive potential of each attribute. The results regarding the connectivity matrix of the first level of abstraction (A) can be found in subsection 3.1.

## 2.4 Complex network measures

Considering the performance and computational cost, the best ML algorithm was used to evaluate the complex network measure's level of abstraction. To characterize the structure of the brain's network, the following complex network measurements were computed as used in the previous work [33–37]: average shortest path length (APL) [93], betweenness centrality (BC) [94], closeness centrality (CC) [95], diameter [96], assortativity coefficient [97, 98], hub score [99], eccentricity [100], eigenvector centrality (EC) [101], average degree of nearest neighbors [102] (Knn), mean degree [103], entropy of the degree distribution [104], transitivity [105, 106], second moment of the degree distribution (SMD) [107], complexity, k-core [108, 109], density [110], and efficiency [111].

In this study, we employed recently developed metrics, as comprehensively detailed in [36], to quantify the number of communities within a complex network. Our investigation also incorporated various community detection algorithms [112–114]. The outcomes of community detection measures were consolidated into a single scalar value. Specifically, we focused on identifying the largest community within each network, followed by the computation of the average path length within that community, resulting in a singular metric. The suite of community detection algorithms encompassed fast greedy (FC) [115], infomap (IC) [116], leading eigenvector (LC) [117], label propagation (LPC) [118], edge betweenness (EBC) [119], spin-glass (SPC) [120], and multilevel community identification (MC) [121]. For clarity and coherence, we extended the abbreviations by appending the letter 'A' (indicating average path length) to denote the corresponding approach, resulting in AFC, AIC, ALC, ALPC, AEBC, ASPC, and AMC.

Further, we used three measures to analyze the segregation and integration concepts: Effective Information (*EI*) and determinism and degeneracy coefficients. Measures of integrated information promise general applicability to questions in neuroscience, in which part-whole relations play a role, and are our interest here [122]. In this paradigm, a system can show a balance between two competing tendencies [123]:

- integration, i.e., the system behaves as one;

- segregation, i.e., the parts of the system behave independently.

In other words, integration in network analysis refers to how well nodes in a network are interconnected, facilitating efficient information flow, and highly integrated networks allow for smooth information exchange between nodes [124, 125]. Segregation, on the other hand, pertains to distinct subgroups or communities within a network; segregated networks have subsets of nodes that are more tightly connected within their subgroups, often forming distinct clusters or communities [126, 127].

The *EI* was first introduced to capture the causal influence between two subsets of neurons as a step in calculating integrated information in the brain [128]. Later, a system-wide version of *EI* was shown to capture fundamental causal properties such as determinism and redundancy [129, 130]. To expand the *EI* framework to networks, in [131], the intervention operation in the *EI* calculation is relaxed by assuming that $W^{\text{out}}$ has modulus one and interpreted as

leaving a random walker on the network. This allows us to investigate the dynamics by inspecting the graph topology. Quantitatively, the *EI* is based on two uncertainties: the first is the Shannon entropy of the average out-weight vector in the network, $H(W_i^{\text{out}})$, which captures how distributed out-weights of the network are; the second is the average entropy of each node's $H(W_i^{\text{out}})$, giving:

$$EI = H(W_i^{\text{out}}) - H(W_i^{\text{out}}).$$

Further, two fundamental components of *EI* are the determinism and degeneracy coefficients. They are based on a network's connectivity, specifically the degree of overlapping weight in the networks. The determinism is based on the average of how much information is not lost in a walker's uncertainty, $H(W_i^{\text{out}})$. Since $\log_2(n)$ represents maximal determinism, i.e., when all walkers have the output $w_{ij} = 1$. Then, the determinism is given by $\log_2(n) - H(W_i^{\text{out}})$. Meanwhile, the degeneracy describes how non-uniform the weight distribution is of the network. If all nodes lead only to one node, that network is perfectly degenerate. The degeneracy can be captured by $\log_2(n) - H(W_i^{\text{out}})$. Together, determinism and degeneracy can be used to re-define *EI*:

$$EI = \text{determinism} - \text{degeneracy}.$$

However, this study considers three classes, differently from the previous ones [33–37] that only consider two classes. Therefore, it was not possible to classify ADHD from TD using the complex network measures. To address this challenge and gain insights into the underlying patterns within the data, we performed a Principal Components Analysis (PCA). PCA is a dimensionality reduction technique that transforms the original high-dimensional data into a lower-dimensional representation while preserving as much of the variance in the data as possible [132, 133]. By extracting the main components, we aimed to uncover hidden structures and reduce the dimensionality of the dataset, which can be beneficial for subsequent analysis and visualization. This approach allowed us to explore the relationships between the variables and potentially reveal patterns that may not be apparent in the raw data, ultimately contributing to a deeper understanding of the complex network measures in the context of ADHD and TD classification.

After PCA, we conducted a statistical analysis using the Wilcoxon test with Bonferroni correction to compare the three classes: ASD, ADHD, and TD. The Bonferroni correction controls the family-wise error rate in multiple hypothesis testing scenarios, such as when performing multiple pairwise comparisons [134]. In our context, it helps address the issue of inflated Type I error rates that can occur when conducting multiple statistical tests simultaneously. The Wilcoxon test, also known as the Mann-Whitney U test in the case of two groups or the Kruskal-Wallis test for more than two groups [135–137], is a non-parametric test used to assess whether there are statistically significant differences between groups when the assumptions of normality and equal variances are not met. In this specific analysis, the Wilcoxon test allowed us to determine if there were significant differences in some measure or variable among the ASD, ADHD, and TD groups. To apply the Bonferroni correction, the significance level (alpha) for each comparison is adjusted to reduce the overall probability of making a Type I error (Type I error, also known as a false positive, occurs when a true null hypothesis is rejected in a statistical test [138]. In the context of multiple comparisons, it refers to the increased likelihood of mistakenly concluding that there is a significant difference when there is not due to the increased number of tests being performed simultaneously. The Bonferroni correction helps to reduce this risk) across all comparisons [139]. This adjustment is achieved by dividing the original alpha level by the number of comparisons being made. The

adjusted alpha becomes more stringent, making it harder to declare a result as statistically significant. Consequently, the Bonferroni correction helps to mitigate the risk of false positives when conducting multiple comparisons. These results can be seen in subsection 3.2, and the following symbols represent the statistical significance:

- ns: $5.00e - 02 < p < = 1.00e + 00$;

- *: $1.00e - 02 < p < = 5.00e - 02$;

- **: $1.00e - 03 < p < = 1.00e - 02$;

- ***: $1.00e - 04 < p < = 1.00e - 03$;

- ****: $p < = 1.00e - 04$.

## 3 Results

### 3.1 Connectivity matrices

According to Table 4, the best classifiers were LSTM and SVM. LSTM performance for the test set was equal to 0.983 for the mean AUC, 0.978 for precision, 0.978 for recall, and 0.978 for accuracy. SVM performance for the test set was equal to 0.946 for the AUC, 0.928 for the precision, 0.928 for the recall, and 0.928 for the accuracy. Each classifier's confusion matrices and ROC curves are depicted in Figs 2 and 3, respectively.

Further, we investigated potential biases arising from variations in data acquisition protocols between different sites to enhance the robustness of our analysis; we compared the TD group for the ADHD dataset and the ASD dataset using the SVM. The results in Fig 4 show that all metric performance stands around 0.50 in a random classifier. Therefore, it is impossible to distinguish between the TD groups of the different datasets, proving that we could mitigate potential biases from variations in data acquisition protocols.

Since SVM has a lower computational cost, it was chosen for the following subsequent analyses. To determine the optimal number of features required for peak performance, we conducted a Recursive Feature Elimination (RFE) analysis, as illustrated in Fig 5A. RFE, often used in the literature in prediction models in medical data [140–142], iteratively removes less

**Table 4. Results from different ML algorithms.** The best ML algorithms were LSTM and SVM, whose performances are highlighted in bold.

| Model | Subset | AUC | Accuracy | Recall | Precision |
|---|---|---|---|---|---|
| LSTM | Train | 0.987±0.022 | 0.968±0.051 | 0.951±0.082 | 0.954±0.074 |
|  | Test | **0.983** | **0.978** | **0.978** | **0.978** |
| SVM | Train | 0.951±0.020 | 0.936±0.026 | 0.936±0.026 | 0.940±0.025 |
|  | Test | **0.946** | **0.928** | **0.928** | **0.928** |
| LR | Train | 0.948±0.022 | 0.931±0.029 | 0.931±0.029 | 0.934±0.028 |
|  | Test | 0.946 | 0.928 | 0.928 | 0.927 |
| CNN | Train | 0.956±0.042 | 0.904±0.048 | 0.852±0.072 | 0.860±0.073 |
|  | Test | 0.938 | 0.920 | 0.913 | 0.913 |
| MLP | Train | 0.932±0.025 | 0.910±0.033 | 0.910±0.033 | 0.913±0.035 |
|  | Test | 0.929 | 0.905 | 0.905 | 0.905 |
| NB | Train | 0.834±0.028 | 0.779±0.037 | 0.779±0.037 | 0.800±0.042 |
|  | Test | 0.854 | 0.805 | 0.805 | 0.812 |

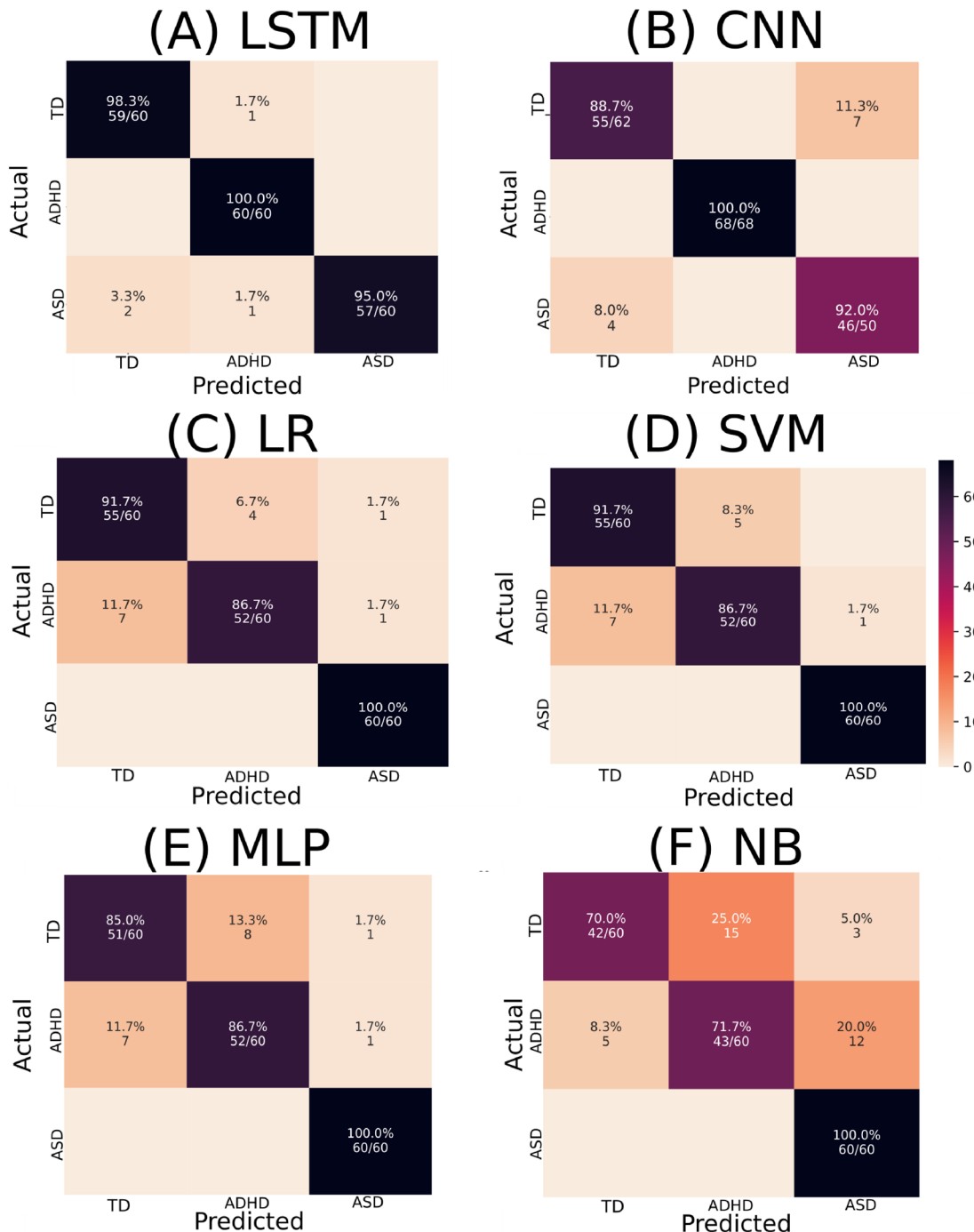

**Fig 2. Confusion matrices depict the performance of various ML algorithms.** The elements in the figure labeled A to F correspond to LSTM, CNN, LR, SVM, MLP, and NB, respectively. The diagonal elements represent TP values, showcasing each algorithm's accuracy in correctly identifying positive instances. This is noteworthy on a test sample containing 180 instances.

critical features to gauge their impact on model performance, allowing us to pinpoint the most relevant features. Fig 5A demonstrates that greater accuracy is attained while using 310 characteristics. Thus, including a complete feature set was not necessary to achieve maximum effectiveness.

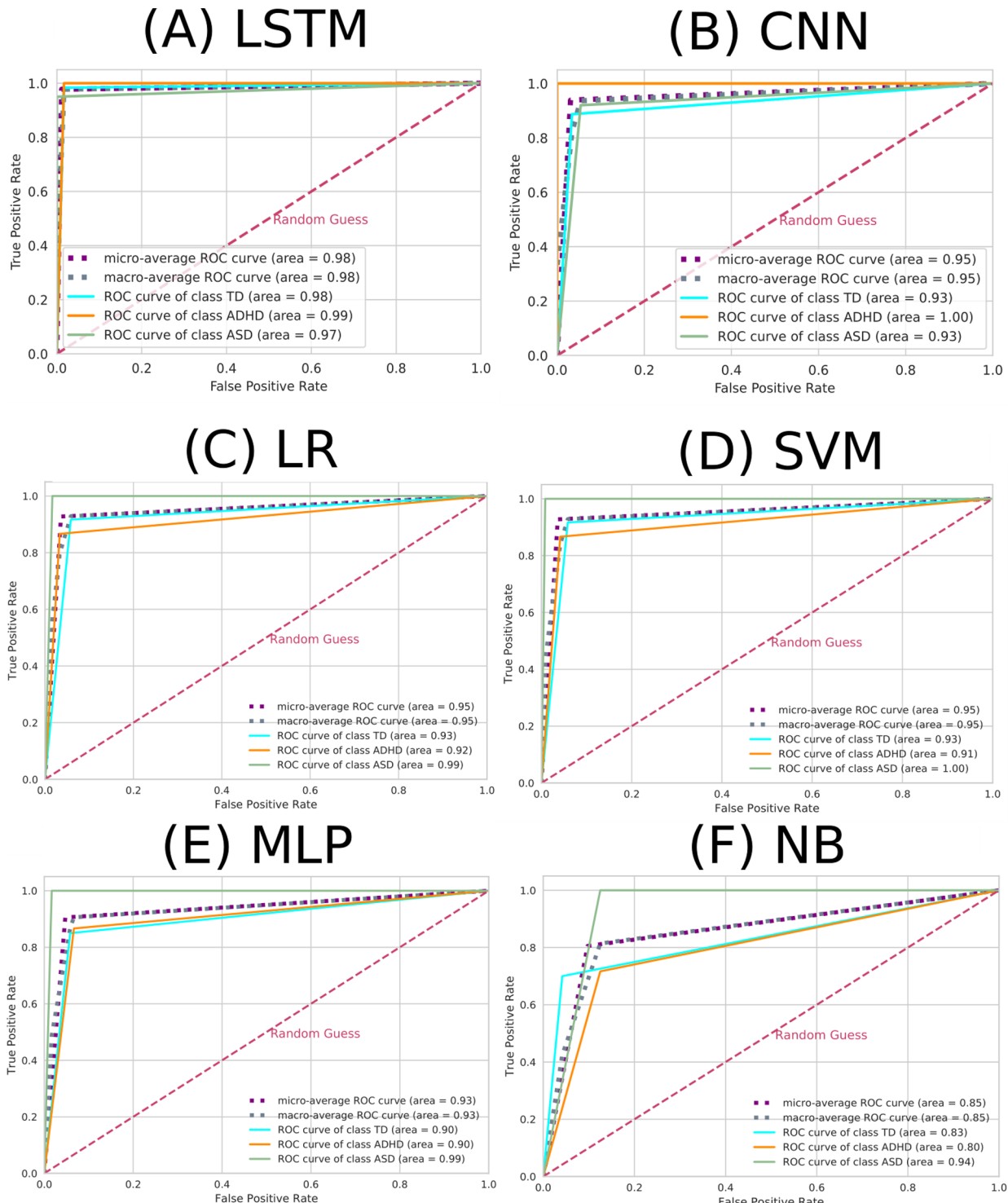

**Fig 3. ROC curve for each ML algorithm.** The elements in the figure labeled A to F correspond to LSTM, CNN, LR, SVM, MLP, and NB, respectively. The dashed pink line represents the random choice classifier, the purple line the micro-average ROC curve, the gray line the macro-average ROC curve, the turquoise line the ROC curve referring to the TD class, the orange line the ROC curve referring to the ADHD class, and the green line the ROC curve referring to the ASD class.

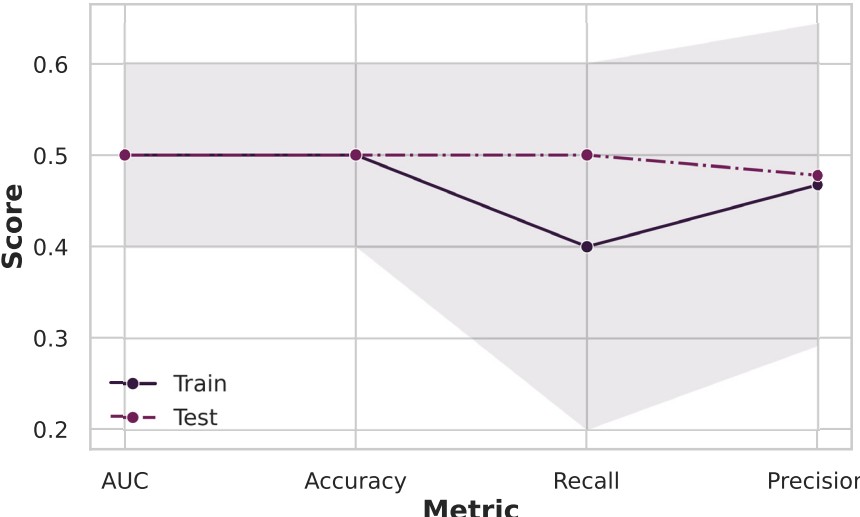

**Fig 4. Investigation of potential biases arising from variations in data acquisition protocols.** Comparing the TD groups from different datasets results in all metric performance standing around 0.50 in a random classifier, proving that we could mitigate potential biases from variations in data acquisition protocols with our preprocessing.

In addition to RFE, we generated a learning curve, as illustrated in Fig 5B, to gain insights into the influence of dataset size on our model's performance. This curve visually represents how the number of training instances affects the model's predictive accuracy. Therefore, RFE and the learning curve enable us to fine-tune our SVM model, ensuring it achieves the optimal equilibrium between feature selection and data volume. From Fig 5B, it can be seen that the maximum performance occurred with 450 subjects without the need for the complete dataset.

Then, we used the 310 features obtained from RFE analysis to perform the SHAP values methodology. The results can be seen in Fig 6.

As in Fig 6, it can be seen that for all classes, mainly for distinguishing TD and ASD (in Fig 6A, dark blue and green, respectively), the two primary connections were Left-ParsOrbitalis-Left-PrimMotor and Left-ParsOrbitalis-Left-Thalamus. Regarding the ASD class, the primary connections found were Left-ParsOrbitalis-Left-Thalamus and Left-ParsOrbitalis-Left-Prim-Motor, with a low correlation value for this class (in Fig 6B). Regarding the ADHD class, the primary connections found were Left-VisualAssoc-Outside defined BAS1 and Right-AngGyrus-Outside defined BAS1, with a low correlation value for this class (in Fig 6C). From our previous work, the Outside defined BAS1 was identified as the cerebellum. Fig 7 contains the two-dimensional schematic (ventral-axis) with the man regions found regarding ASD and ADHD.

Furthermore, we introduced noise generated by a normal distribution, with different means (level of the noise) while keeping the standard deviation constant at 0.1. This resulted in a range of noisy matrices that we used to evaluate our SVM model's performance. We assessed the SVM model's performance using AUC and accuracy depicted in Fig 8, which indicates that our SVM model exhibits robustness to noise in the input data matrices. Even when noise levels vary from 0.0 to 1.5, the model maintains a relatively high AUC and accuracy, indicating its ability to accurately classify patients with ASD, ADHD, and TD individuals. The model's performance gradually decreases as the noise level increases, which is expected. However, it is noteworthy that even at noise levels as high as 1.0 (where the data is significantly distorted), the model still achieves a reasonable AUC of 0.70 and an accuracy of 0.60 (see Fig 8).

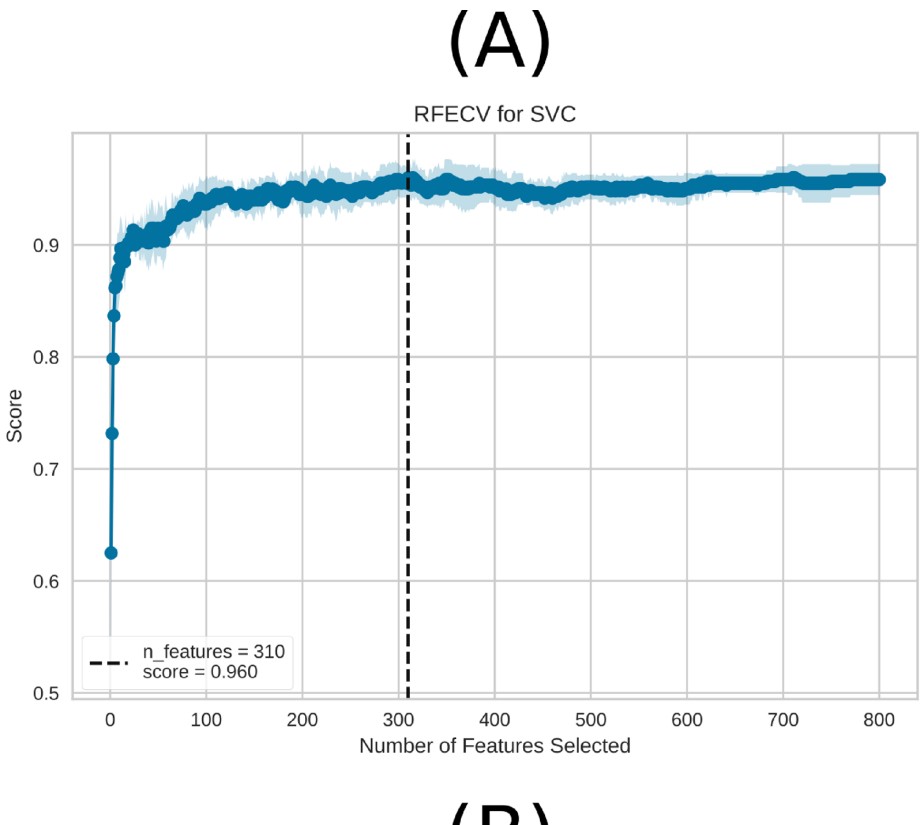

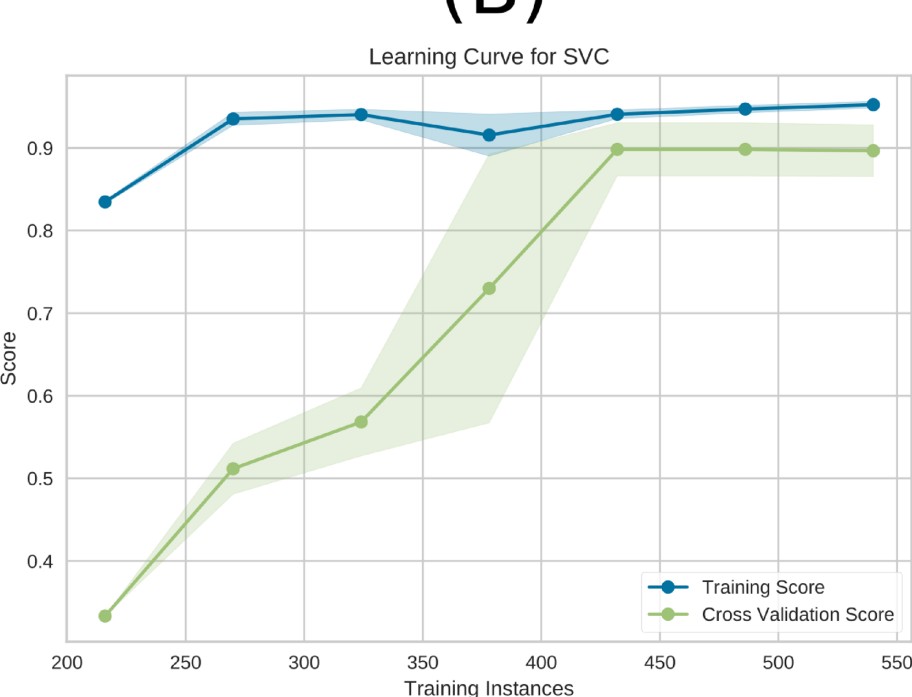

**Fig 5. RFE and the learning curve for the SVM model are depicted in (A) and (B), respectively.** The best performance is achieved with a total of 310 characteristics, as shown by (A). In (B), the learning curve is presented for the training Accuracy (blue) and test Accuracy (green) using the entire dataset (600 connectivity matrices subjects). The highest performance was achieved with 450 connectivity matrices subjects without requiring the entire dataset. Notably, the connectivity matrices were generated using the data augmentation technique sliding window, and 600 connectivity matrices were used in total in the ML approach before the sampling technique.

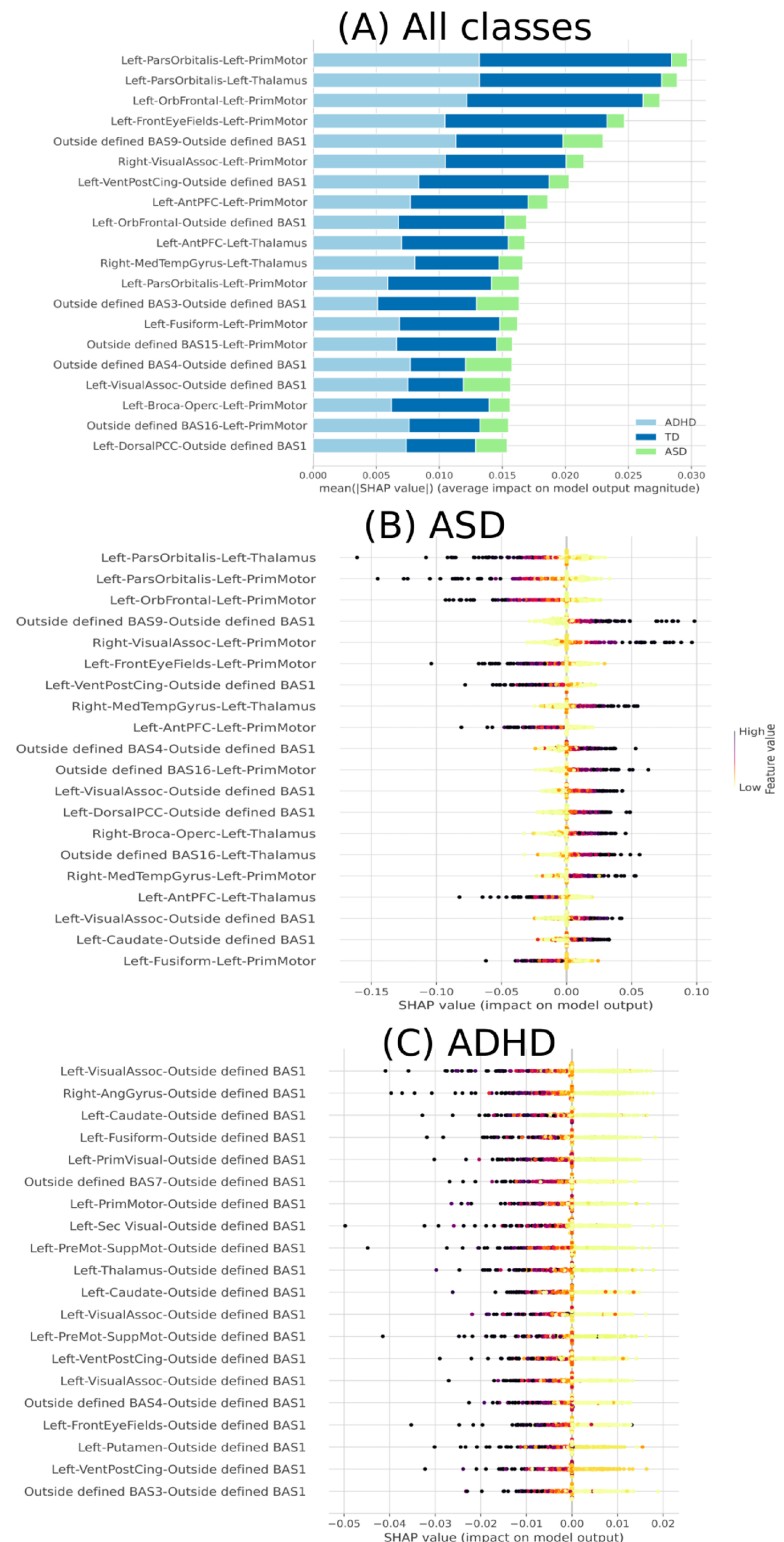

**Fig 6. Feature importance ranking using the SHAP values methodology for the SVM classifier with brain regions in descending order.** (A) Feature ranking based on the average of absolute SHAP values over all subjects considered. (B) Feature importance ranking regarding ASD class. (C) Feature importance ranking regarding ADHD class.

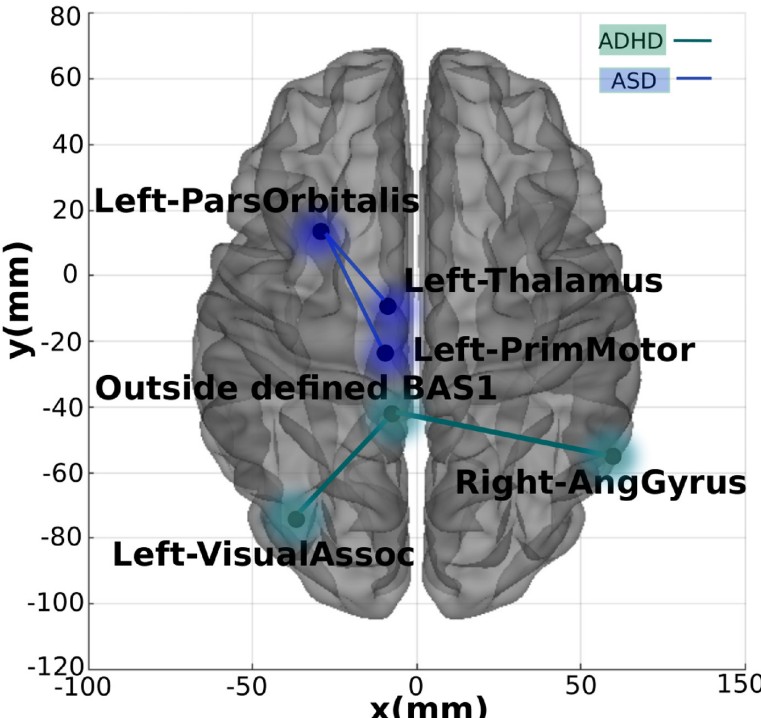

**Fig 7. The most important connections found.** Two-dimensional schematic (ventral-axis), where the most important connection for ADHD and ASD are highlighted in green and blue, respectively. The brain plot was developed by the Braph tool [143], and each region was plotted using a Brodmann map from the Yale BioImage Suite Package.

This suggests that the SVM model is resilient to noise and can provide valuable diagnostic information in real-world scenarios where data may be imperfect.

Further, we conducted stratified k-fold cross-validation with values beyond k = 10, namely 2, 3, 5, and 15. The resulting plot in Fig 9 reveals trends in the SVM model's performance on

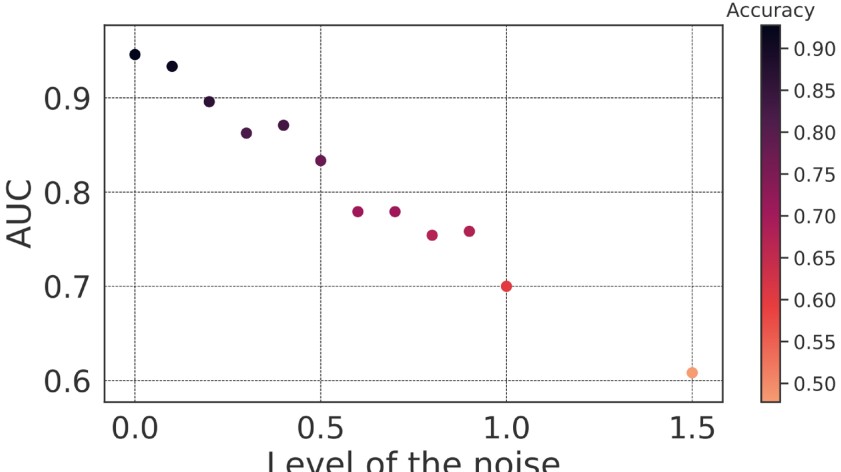

**Fig 8. SVM behavior after insertion of noise.** The mean AUC of the test was obtained with the insertion of noise generated by a normal distribution with 0.1 standard deviation and a 0–1.5 mean range.

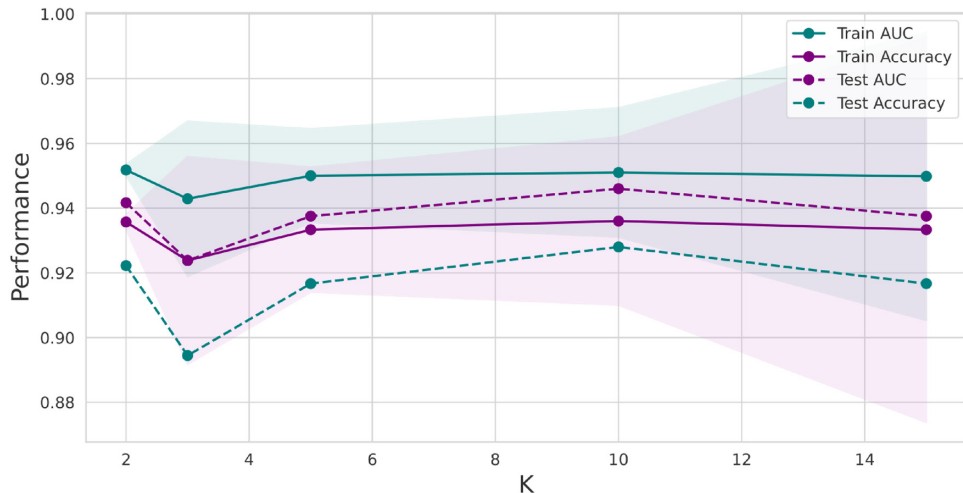

**Fig 9. Plot for the SVM model with performance measures.** The AUC and Accuracy, in the y-axis, in blue and purple, respectively, were obtained by varying the number of k in the stratified k-fold-cross-validation (x-axis))—the dashed lines corresponding to the test sample and the complete lines to the training sample. Furthermore, the shaded represents the standard deviation in the training sample.

the training dataset. As the value of k increases, the AUC remains relatively stable, with values ranging from approximately 0.94 to 0.95. This suggests that the SVM model consistently discriminates between the three groups—ASD, ADHD, and TD. The corresponding accuracy values remain steady, ranging from approximately 0.93 to 0.94. We observed similar stability in AUC values for the test dataset, which ranges from about 0.92 to 0.95 as k varies. This indicates that the SVM model's ability to distinguish between groups holds when applied to unseen data. The accuracy on the test dataset also remains steady, with values ranging from approximately 0.89 to 0.93. This robustness is particularly valuable when dealing with real-world data where k can impact model stability.

## 3.2 Complex network

The performance of the test sample considering the complex network yielded the confusion matrix and the ROC curve depicted in Fig 10.

The performance of the test sample considering the complex network yielded the confusion matrix, and the ROC curve depicted in Fig 10 indicates that the model did not perform well for the ADHD class (with an accurate positive accuracy of 0.53 and an AUC of 0.72). This suboptimal performance can be attributed to several factors. Firstly, when we performed PCA with two and three components, as shown in Fig 11, it became evident that the ADHD and TD instances formed two overlapping groups, differently from the ASD instance class. This lack of clear separation in the PCA space suggests that the initial feature set does not easily capture the inherent characteristics distinguishing ADHD from TD cases. This inherent overlap in feature distributions can significantly hinder the performance of a classifier like SVM, which relies on well-defined class boundaries.

Additionally, we observed in Fig 11 that none of the features displayed strong correlations with the principal components. This lack of feature-component solid correlations suggests that the initial feature set may need to contain clear discriminatory information, making it challenging for the SVM to distinguish between classes effectively.

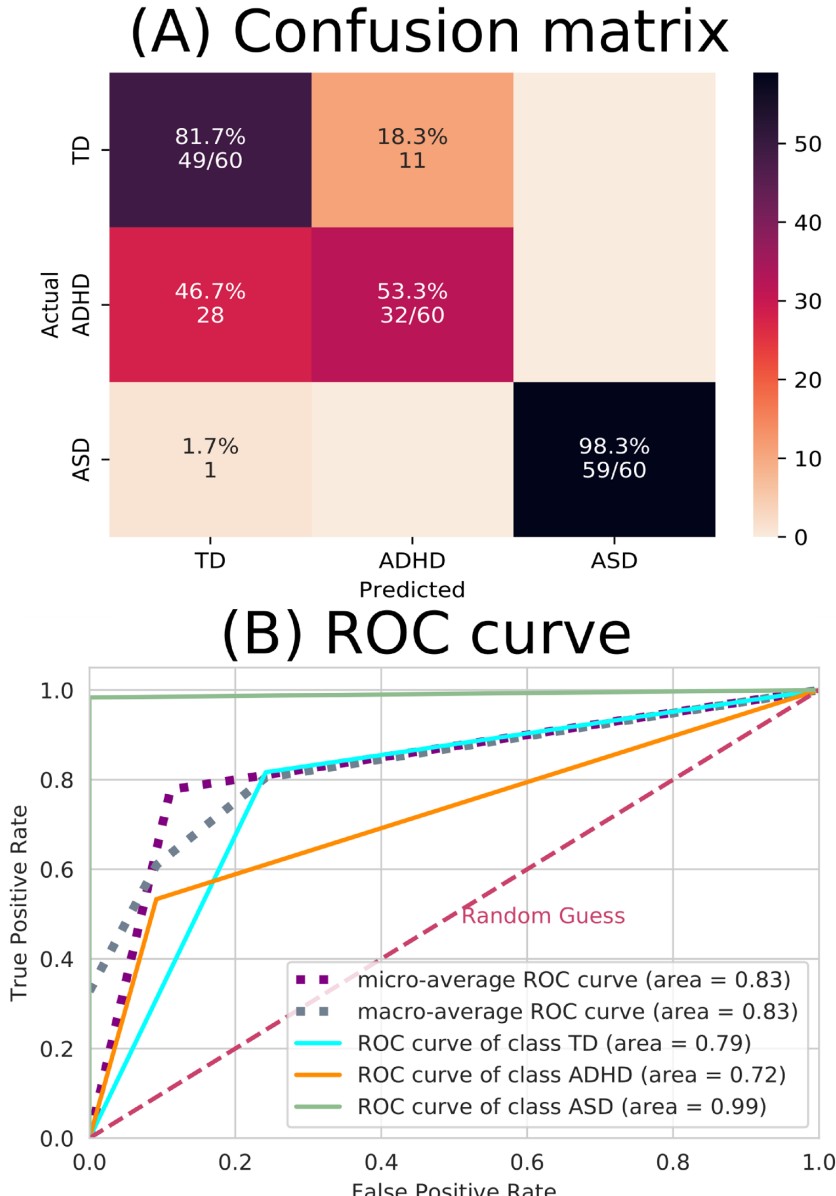

**Fig 10. ML results from complex network measures.** (A) The confusion matrix indicates that there were a lot of incorrect predictions between the TD and ADHD groups. (B) The ROC curve, where the dashed pink line represents the random choice classifier, the purple line is the micro-average ROC curve, the gray line is the macro-average ROC curve, the turquoise line the ROC curve referring to the TD class, the orange line the ROC curve referring to the ADHD class (which can be seen the ADHD has the lowest-distinguished curve) and the green line the ROC curve referring to the ASD class (which can be seen the ASD has the best-distinguished curve).

Therefore, to improve the model performance, it may be necessary to consider additional domain-specific features that could better capture the nuances of ADHD and TD differentiation within the dataset.

Then, we performed a statistical t-test with Bonferroni correction. This choice was driven by our need to rigorously assess the significance of differences in the means of individual features between the ADHD and TD groups. By conducting this test, we could identify which

# (A) PCA with 3 components

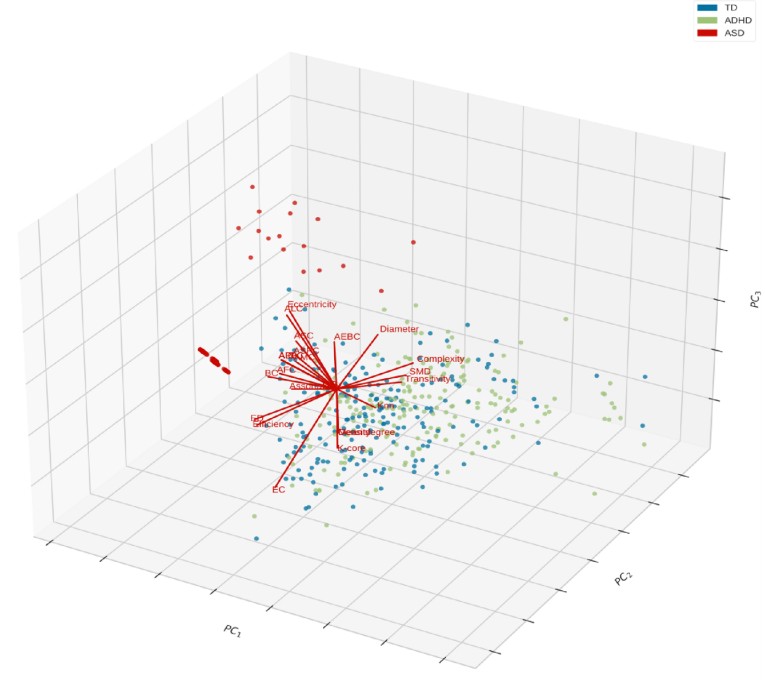

# (B) PCA with 2 components

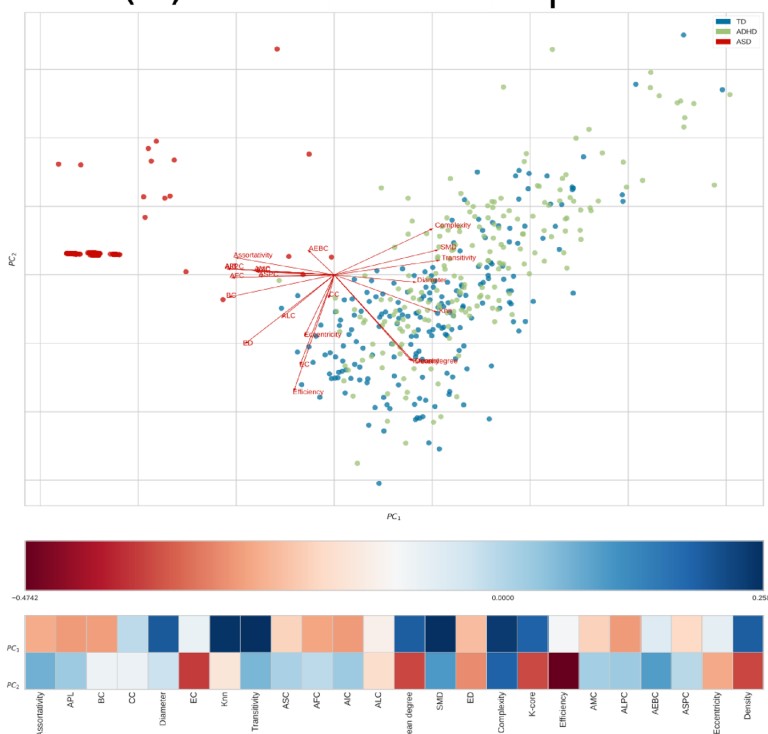

**Fig 11. PCA using the complex network measures.** The features for ASD, ADHD, and TD are depicted in red, green, and blue, respectively. In (A), PCA with three components, namely PC1, PC2, and PC3, is illustrated in the plot axis. In (B), PCA with two components, namely PC1 and PC2, is presented in the plot axis; further, the heatmap shows that any of the features were highly correlated with the two components.

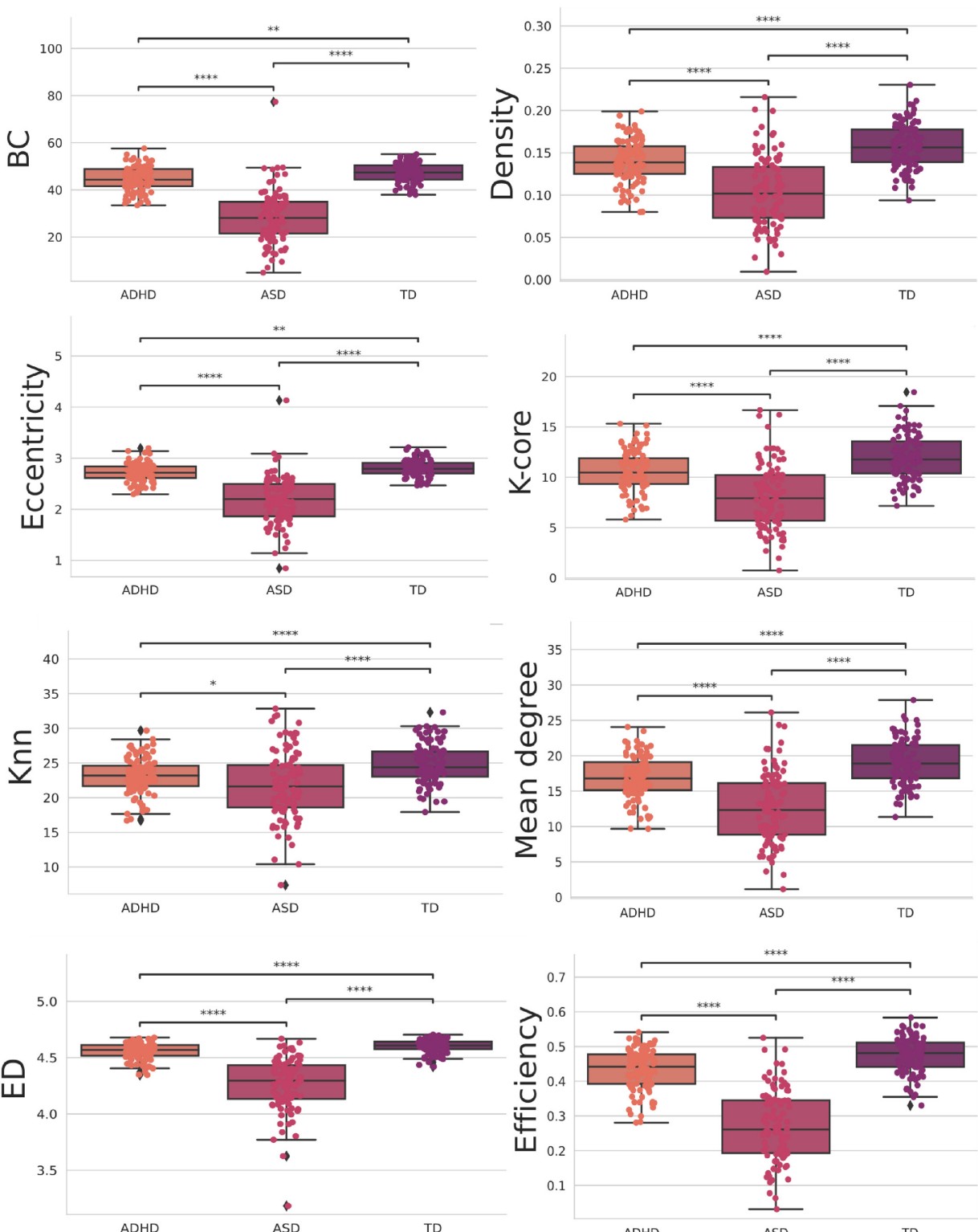

**Fig 12. Features were statistically significant between all groups when using the t-test with Bonferroni correction.** The orange, pink, and purple boxplots show the features that obtained the most statistically significant differences regarding the classes ADH, ASD, and TD, respectively.

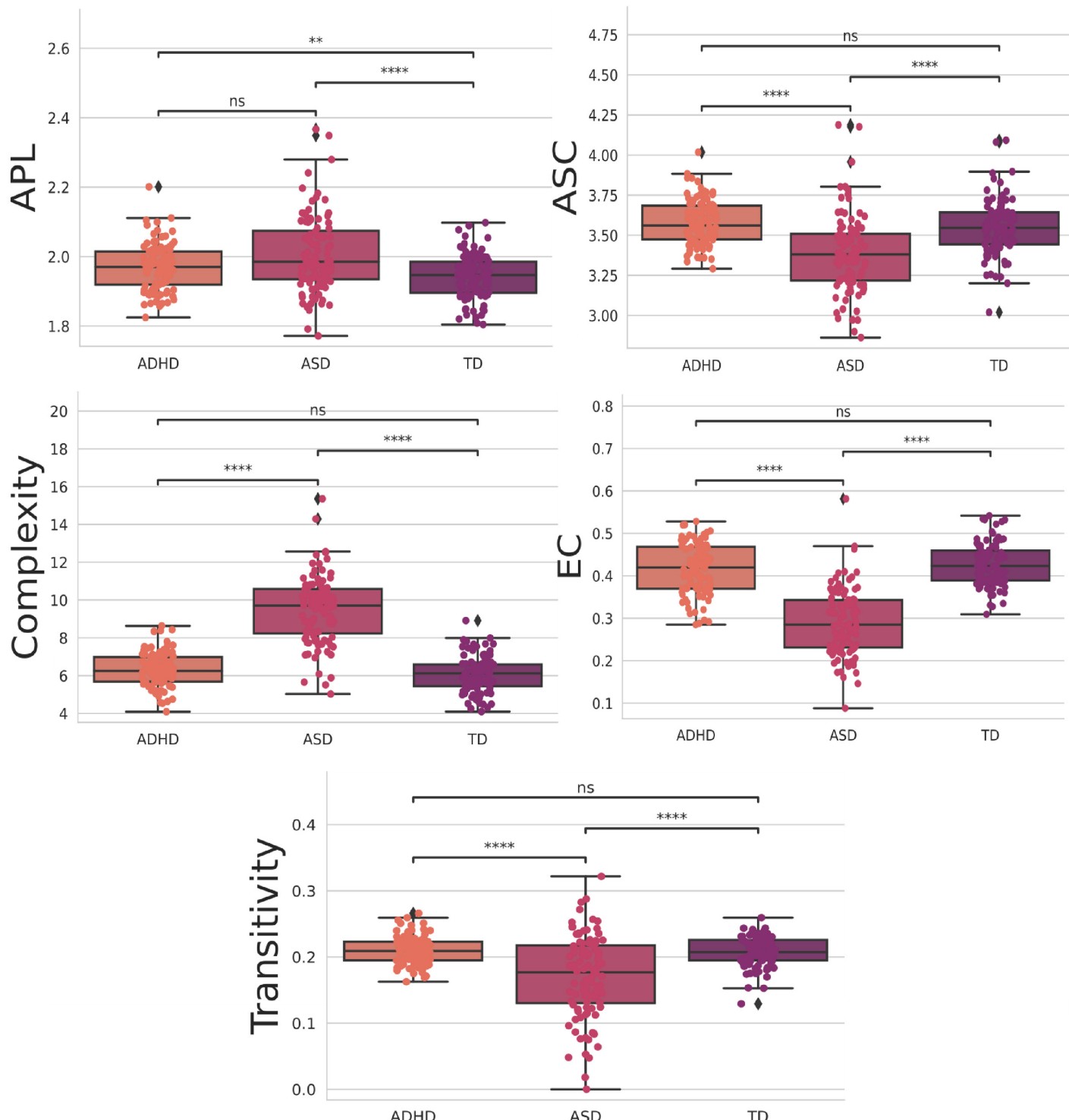

**Fig 13. Features were four stars statistically significant, at least between one of the groups, when using the t-test with Bonferroni correction.** The orange, pink, and purple boxplots show the features that obtained the most statistically significant differences regarding the classes ADH, ASD, and TD, respectively.

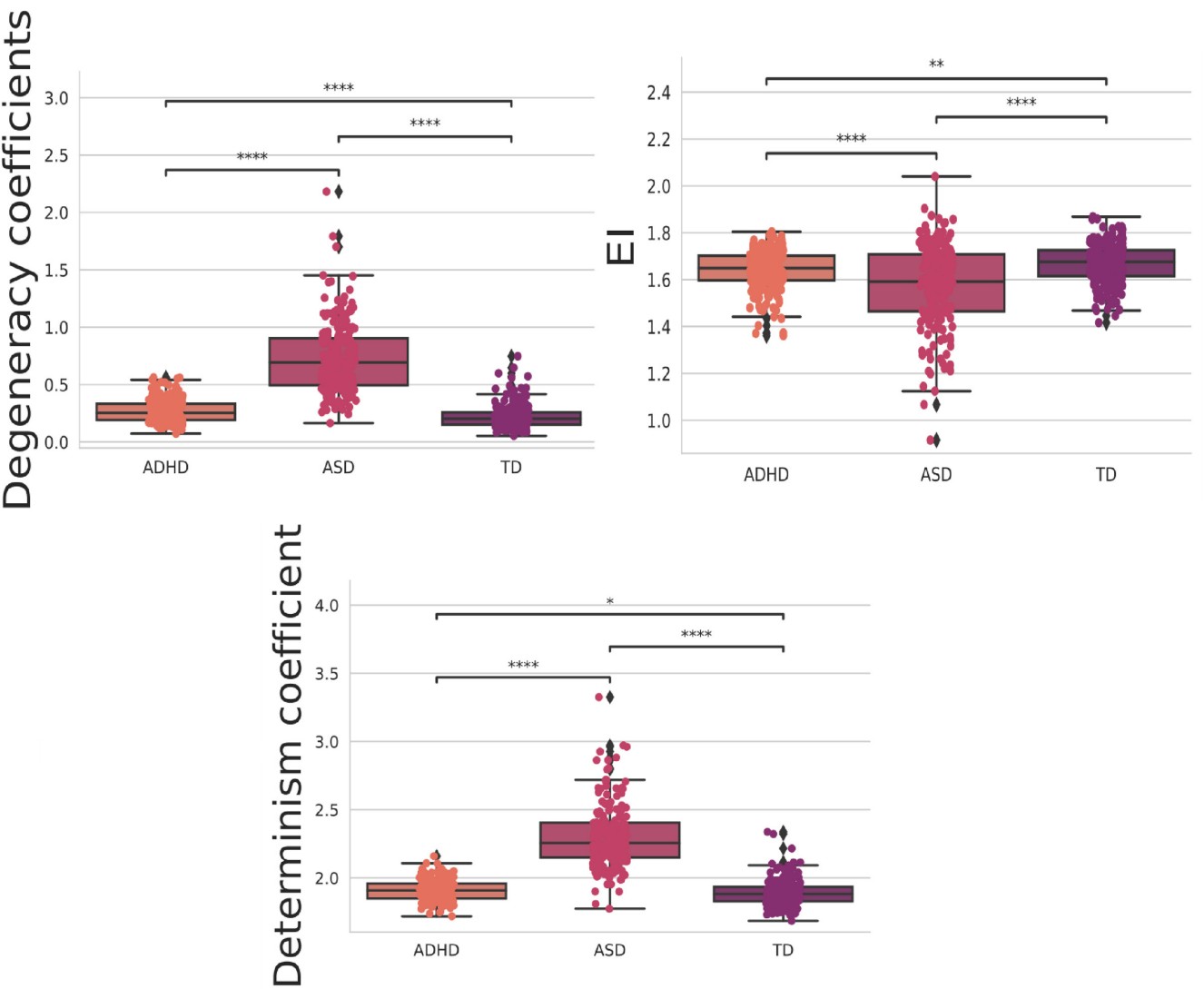

**Fig 14. The t-test with Bonferroni correction for the integrated measures.** The orange, pink, and purple boxplots show the features that obtained the most statistically significant differences regarding the classes ADH, ASD, and TD, respectively.

specific features exhibit statistically significant distinctions between these two classes. The Bonferroni correction is applied to mitigate the issue of multiple comparisons, ensuring that we maintained a low family-wise error rate. In other words, it helps control the higher probability of obtaining false positives when examining numerous features simultaneously. The statistically significant ones between all groups are depicted in Fig 12, and the ones with four stars at least between one of the groups are in Fig 13. Further, the statistical test with the integrated measures can be found in Fig 14.

## 4 Discussion

### 4.1 Connectivity matrices

Overall, we obtained the best performance compared to the multiclass machine learning algorithm comparing ASD, ADHD, and TD in the literature, as described in section 1. Analysis

from Table 1 reveals that our methodology outperforms existing multiclass approaches. In our prior research [33] focusing on EEG time series, we demonstrated the superior accuracy of constructing connectivity matrices compared to conventional methods employing raw EEG data. This underscores the significance of network topology in characterizing brain data.

Furthermore, in subsequent investigations [34, 35], we found that employing a distinct correlation metric yielded improved detection of brain changes associated with ASD and schizophrenia, respectively. Interestingly, TE proved effective in capturing such changes in the fMRI dataset. Thus, one of our hypotheses for achieving optimal performance revolves around selecting an appropriate correlation metric.

Furthermore, our findings, as illustrated in Fig 6, provide valuable insights into the functional roles of specific brain regions in distinguishing between individuals with ASD, ADHD, and TD based on fMRI matrices. These results shed light on the neural circuitry implicated in these neurodevelopmental conditions.

In Fig 6A, where TD and ASD are distinguished, the two primary connections of interest are Left-ParsOrbitalis-Left-PrimMotor and Left-ParsOrbitalis-Left-Thalamus. This observation suggests that these connections play a significant role in discriminating between TD and ASD individuals. The Left-ParsOrbitalis is associated with decision-making and social cognition [144], areas commonly affected in individuals with ASD [145]. The connections to the PrimMotor and Thalamus imply that motor control and sensory processing also contribute to distinguishing between these groups and are also found in our previous work [34]. Further, Fig 6B highlights the primary connections for the ASD class, with a focus on Left-ParsOrbitalis-Left-Thalamus and Left-ParsOrbitalis-Left-PrimMotor. These connections are consistent with the findings in Fig 6A, emphasizing the importance of the Left-ParsOrbitalis in distinguishing individuals with ASD. This region's involvement in social cognition, decision-making, and language processing may reflect the cognitive and behavioral characteristics associated with ASD.

In Fig 6C, which pertains to the ADHD class, the primary connections identified are Left-VisualAssoc-Outside defined BAS1 and Right-AngGyrus-Outside defined BAS1, both with low correlation values. Our previous work has identified the "Outside defined BAS1" as the cerebellum. The cerebellum is traditionally linked to motor control [146]. However, emerging research suggests its involvement in cognitive functions, including attention and executive control [147] and in several studies associated with ADHD [146, 148, 149]. The Left-VisualAssoc connection might signify differences in visual processing [150], while the Right-AngGyrus could be related to higher-order cognitive functions [151]. Both areas play a significant role in visuospatial attention processes [152, 153]. The low correlation values may imply that these connections are less distinctive for distinguishing ADHD from the other groups, suggesting a more complex neural signature for this condition.

## 4.2 Complex network measures

The results in Fig 12 offer valuable insights into the network properties of three distinct groups: ASD, ADHD, and TD. We employed a range of complex network measures to assess valuable insights into the network properties of three distinct groups. Further, with these metrics, we can observe distinct patterns in integration and segregation, which are fundamental concepts in network analysis, across the three groups [154, 155].

The ASD group exhibits the lowest values across various network metrics, such as BC, Density, Eccentricity, K-core, KNN, Mean Degree, ED, and Efficiency. This suggests that individuals with ASD have a more fragmented and segregated network structure, indicating challenges in information integration within the network. In other words, their networks may have more

isolated clusters of nodes that do not communicate effectively with each other, according to the literature [34, 156, 157].

On the other hand, the ADHD group demonstrates higher values in these metrics compared to ASD; however, it falls short of the performance observed in the TD group, indicating that individuals with ADHD have a network structure that is more integrated than ASD but still not as optimal as the control group. This implies that individuals with ADHD have a network structure that is more cohesive than ASD but not as cohesive as typical development.

Furthermore, as shown in Fig 13, the ASD group exhibits the lowest values for both the ASC measure, gauging the size of community networks, EC, network influence, and transitivity. In contrast, the ADHD group displays higher values in these metrics than the ASD group but still lags behind the TD group. These findings suggest that individuals with ASD may have smaller and less influential networks within their communities, while those with ADHD fall in between ASD and TD individuals regarding these network characteristics.

By computing the *EI* in the TD, ASD, and ADHD groups, there is a lower average value of *EI* in the ASD group than the others, showing a tendency to have a more segregated than integrated structure in this group (see Fig 14). Interestingly, there is an analysis of determinism and degeneracy coefficients in the groups. The greater value of determinism and degeneracy coefficients in the ASD than the others shows that the graph structure in this group resembles a star (sparse connections) instead of a complete (well-connected) network [131] strengthening the *EI* interpretation in the last paragraph.

In [158], functional segregation was characterized as the capacity for specialized processing within tightly interconnected brain regions. In other words, neuronal processing is distributed across functionally related regions organized into modules. These modules are described as communities exhibiting dense internal connectivity among their constituent nodes and limited communication with nodes from other communities. This network analysis can be linked to the long-observed fact by clinicians that those with ASD are impaired in their ability to generalize—that is, to relate new stimuli to past experiences. Instead, these groups are good at specializing in learned habits [159, 160].

## 5 Conclusion

Our study used fMRI datasets and explainable IA methods to generate an interpretable classifier for ASD, ADHD, and TD. We have found distinct brain activity patterns underlying these neurodevelopmental disorders by advancing beyond binary comparisons and integrating complex network measures alongside machine learning methodologies.

Our findings confirm the existence of unique neural signatures for ASD, ADHD, and TD groups. Notably, connections involving Left-ParsOrbitalis emerged as crucial in distinguishing between TD and ASD, possibly indicating underlying deficits in decision-making and social cognition observed in ASD. Similarly, distinct neural signatures were observed for ADHD, with connections to the cerebellum, Left-VisualAssoc, and Right-AngGyrus, highlighting potential involvement in cognitive functions and sensory processing differences. The observed connectivity patterns on which the ML classification rests agree with established diagnostic approaches based on clinical symptoms, proving the trustworthiness and efficiency of our multiclass ML approach's interpretability technique. Moreover, we demonstrate the superior performance of our multiclass machine learning approach compared to existing literature. This heightened performance is essential for reliable discrimination between neurodevelopmental conditions, promising prospects for more precise diagnostic tools.

Furthermore, our analysis of complex network measures elucidated the network properties of each group, unveiling differences in integration and segregation patterns. The ASD group

exhibited the lowest values across various network metrics, suggesting a fragmented network structure. In contrast, the ADHD group demonstrated intermediate values, indicative of a network that is more integrated than ASD but less cohesive than typical development.

Despite these significant contributions, limitations such as data quantity constrain the generalizability of our findings. Future studies should aim to overcome these limitations by incorporating larger datasets encompassing a broader range of mental health conditions. Further investigations focusing on specific brain regions could provide deeper insights into group differences in brain connectivity. Further, we propose integrating our methodology with federated learning techniques as a promising avenue for advancing diagnostics and drug trials in neurodevelopmental conditions [161–163]. Federated learning offers a solution to data privacy and scalability challenges, allowing for collaborative model training across multiple datasets while preserving data decentralization [164–166]. This approach holds immense potential for improving diagnostic accuracy and guiding personalized treatment strategies tailored to specific demographics or clinical settings [167].

In summary, our study represents a significant step forward in understanding the neural underpinnings of neurodevelopmental conditions. By leveraging advanced analytical techniques and machine learning methodologies, we have surpassed performance in discrimination between ASD, ADHD, and TD individuals, paving the way for refined diagnostics and promising avenues for developing trustworthy clinical decision-support systems.

## Supporting information

**S1 File. Connectivity matrices attached in a zip file used to generate the results.** This section contains the normalized transfer entropy connectivity matrices derived from the authors' preprocessed BOLD time series of 122 fMRI regions. These matrices served as inputs for the algorithms employed in this study and for extracting complex network measures.
(ZIP)

**S2 File.**
(ZIP)

## Author Contributions

**Conceptualization:** Caroline L. Alves.

**Formal analysis:** Caroline L. Alves, Tiago Martinelli.

**Funding acquisition:** Michael Moeckel.

**Investigation:** Caroline L. Alves, Tiago Martinelli.

**Methodology:** Caroline L. Alves.

**Project administration:** Michael Moeckel.

**Resources:** Michael Moeckel.

**Software:** Caroline L. Alves, Tiago Martinelli.

**Supervision:** Michael Moeckel.

**Validation:** Caroline L. Alves, Tiago Martinelli, Loriz Francisco Sallum, Francisco Aparecido Rodrigues, Thaise G. L. de O. Toutain, Joel Augusto Moura Porto, Christiane Thielemann, Patrícia Maria de Carvalho Aguiar, Michael Moeckel.

**Visualization:** Caroline L. Alves.

**Writing – original draft:** Caroline L. Alves, Tiago Martinelli, Loriz Francisco Sallum.

**Writing – review & editing:** Caroline L. Alves, Francisco Aparecido Rodrigues, Thaise G. L. de O. Toutain, Joel Augusto Moura Porto, Christiane Thielemann, Patrícia Maria de Carvalho Aguiar, Michael Moeckel.

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
