## [Decision Letter · Decision Letter 0]

15 Feb 2024

PONE-D-23-39121Multiclass machine learning and deep learning for analysis of fMRI functional connectivity from individuals with Autism Spectrum Disorder and Attention Deficit Hyperactivity DisorderPLOS ONE

Dear Dr. Alves,

Thank you for submitting your manuscript to PLOS ONE. After careful consideration, we feel that it has merit but does not fully meet PLOS ONE’s publication criteria as it currently stands. Therefore, we invite you to submit a revised version of the manuscript that addresses the points raised during the review process.

We look forward to receiving your revised manuscript.

Kind regards,

Yangsong Zhang, Ph.D.

Academic Editor

PLOS ONE

Journal Requirements:

Whilst you may use any professional scientific editing service of your choice, PLOS has partnered with both American Journal Experts (AJE) and Editage to provide discounted services to PLOS authors. Both organizations have experience helping authors meet PLOS guidelines and can provide language editing, translation, manuscript formatting, and figure formatting to ensure your manuscript meets our submission guidelines. To take advantage of our partnership with AJE, visit the AJE website (http://aje.com/go/plos) for a 15% discount off AJE services. To take advantage of our partnership with Editage, visit the Editage website (www.editage.com) and enter referral code PLOSEDIT for a 15% discount off Editage services. If the PLOS editorial team finds any language issues in text that either AJE or Editage has edited, the service provider will re-edit the text for free.

EpilabKI is funded through the Bavarian stated Ministery for Sciences and the Arts research.

EpilabKI is funded through the Bavarian stated Ministery for Sciences and the Arts research.

We note that one or more of the authors are employed by a commercial company.

“The funder provided support in the form of salaries for authors, but did not have any additional role in the study design, data collection and analysis, decision to publish, or preparation of the manuscript. The specific roles of these authors are articulated in the ‘author contributions’ section.”

7. PLOS requires an ORCID iD for the corresponding author in Editorial Manager on papers submitted after December 6th, 2016. Please ensure that you have an ORCID iD and that it is validated in Editorial Manager. To do this, go to ‘Update my Information’ (in the upper left-hand corner of the main menu), and click on the Fetch/Validate link next to the ORCID field. This will take you to the ORCID site and allow you to create a new iD or authenticate a pre-existing iD in Editorial Manager. Please see the following video for instructions on linking an ORCID iD to your Editorial Manager account: https://www.youtube.com/watch?v=_xcclfuvtxQ

Additional Editor Comments:

The authorship should address the issues from the two experts before the further consideration.

Reviewers' comments:

Reviewer's Responses to Questions

**Comments to the Author**

1. Is the manuscript technically sound, and do the data support the conclusions?

Reviewer #1: Partly

Reviewer #2: Yes

2. Has the statistical analysis been performed appropriately and rigorously? 

Reviewer #1: No

Reviewer #2: No

3. Have the authors made all data underlying the findings in their manuscript fully available?

Reviewer #1: Yes

Reviewer #2: Yes

4. Is the manuscript presented in an intelligible fashion and written in standard English?

Reviewer #1: Yes

Reviewer #2: No

5. Review Comments to the Author

Reviewer #1: The authors used multiclass machine learning and deep learning algorithms to distinguish ASD, ADHD, and TD. The study has a lot of major issues, which I will outline below following the order of the manuscript:

1. The title does not accurately represent their current work. Since the present study was focused on the classification of ASD, ADHD, and TD, there should be more descriptions involving classification.

2. The Abstract of this manuscript is excessively redundant and lacks a concise summary of the article's innovation. The authors should try to rewrite this section by following a specific order and to make it more readable for the readers.

3. The Introduction needs to be roundly restructured so that it clearly reflects the hypothesis, questions, and aim of the study. Besides, in the fifth and seventh paragraphs of the Introduction, necessary contents and explanations are lacking.

4. It seems that the neuroimaging data of ASD and ADHD were obtained from two independent datasets, how did the authors eliminate the effects between different sites?

5. More explanations are needed for the choice of time window. With the current description, it is not known how their network is constructed. I suggest the authors make an effort to develop the Sections, giving more notions on the network generation and the related deep learning algorithms.

6. Likewise, in the section Connectivity matrices, some necessary descriptions are missing.

7. This submission seems to be a mere application of machine learning and deep learning algorithms to ASD and ADHD, the authors should point out clearly the degree of methodological novelty of the current submission w.r.t. the previous work.

8. A brief comparison with the state-of-the-art can be included in the discussion section to highlight the impact of the study.

9. The discussion was too sample, I think there should be more discussion about the physiological meaning of the results and why the current method achieved better classification performance.

10. Please discuss future directions regarding diagnostics and drug trials.

11. Some English typos were noticed, the authors should carefully and thoroughly check the English writing to avoid these typos.

Reviewer #2: First of all, this paper made three classifications on ASD, ADHD and TC, and carried out corresponding analysis on the two diseases. A good classification effect was achieved based on the used data, and the corresponding analysis was carried out. However, the paper still had the following shortcomings:

Disadvantages:

1. The amount of data used is mentioned in the abstract, but it is not clear whether there are 40 subjects in three categories respectively or 40 subjects in three categories altogether.

2. The amount of data used is too small, and the pre-processing part of the data should be detailed.

3. Lack of innovation, less workload, can appropriately increase the amount of data to carry out experiments.

4.LSTM has significantly better performance than SVM, so why not try LSTM for analysis instead of SVM for reducing computation?

5. In Figure 4-b, it is necessary to explain where many subjects come from and how to deal with them.

6.The paper explains the use of transfer entropy to calculate the functional network, and the specific calculation formula can be written as much as possible in this place, and there is also a question that your connectivity matrix is symmetrical in Figure 1, whether it conforms to the connectivity matrix of transfer entropy calculation.

6. PLOS authors have the option to publish the peer review history of their article (what does this mean?). If published, this will include your full peer review and any attached files.

Reviewer #1: No

Reviewer #2: No

---

## [Author Response · Author response to Decision Letter 0]

14 Apr 2024

Dear all,

We would like to thank you, the reviewer, for carefully reading our manuscript. The comments have improved

the manuscript. We provide point-by-point responses to each comment below, which are attached to the author response pdf. 

Best regards,

Authors

---

## [Decision Letter · Decision Letter 1]

29 May 2024

PONE-D-23-39121R1Multiclass Classification of Autism Spectrum Disorder, Attention Deficit Hyperactivity Disorder, and Typically Developed Individuals Using fMRI Functional Connectivity AnalysisPLOS ONE

Dear Dr. Alves,

Thank you for submitting your manuscript to PLOS ONE. After careful consideration, we feel that it has merit but does not fully meet PLOS ONE’s publication criteria as it currently stands. Therefore, we invite you to submit a revised version of the manuscript that addresses the points raised during the review process.

We look forward to receiving your revised manuscript.

Kind regards,

Yangsong Zhang, Ph.D.

Academic Editor

PLOS ONE

Journal Requirements:

Additional Editor Comments:

Some minor issues should be addressed.

Reviewers' comments:

Reviewer's Responses to Questions

**Comments to the Author**

1. If the authors have adequately addressed your comments raised in a previous round of review and you feel that this manuscript is now acceptable for publication, you may indicate that here to bypass the “Comments to the Author” section, enter your conflict of interest statement in the “Confidential to Editor” section, and submit your "Accept" recommendation.

Reviewer #1: All comments have been addressed

Reviewer #2: All comments have been addressed

2. Is the manuscript technically sound, and do the data support the conclusions?

Reviewer #1: Yes

Reviewer #2: Yes

3. Has the statistical analysis been performed appropriately and rigorously? 

Reviewer #1: Yes

Reviewer #2: Yes

4. Have the authors made all data underlying the findings in their manuscript fully available?

Reviewer #1: Yes

Reviewer #2: Yes

5. Is the manuscript presented in an intelligible fashion and written in standard English?

Reviewer #1: Yes

Reviewer #2: Yes

6. Review Comments to the Author

Reviewer #1: (No Response)

Reviewer #2: In the chapter 2.2 Data and data preprocessing, the spelling of "ABIDE" is missing "A"(page 5). You should pay attention to these spelling problems and check the words in the whole text as well as sentence problems.

7. PLOS authors have the option to publish the peer review history of their article (what does this mean?). If published, this will include your full peer review and any attached files.

Reviewer #1: No

Reviewer #2: No

---

## [Author Response · Author response to Decision Letter 1]

1 Jun 2024

We would like to thank you, the reviewer, for carefully reading our manuscript. The comments have improved the manuscript. The response to the reviewers can be found in the attached file named: "Author_response_to_review-version2.pdf"

---

## [Editor Report · Decision Letter 2]

4 Jun 2024

Multiclass Classification of Autism Spectrum Disorder, Attention Deficit Hyperactivity Disorder, and Typically Developed Individuals Using fMRI Functional Connectivity Analysis

PONE-D-23-39121R2

Dear Dr. Alves,

We’re pleased to inform you that your manuscript has been judged scientifically suitable for publication and will be formally accepted for publication once it meets all outstanding technical requirements.

Kind regards,

Yangsong Zhang, Ph.D.

Academic Editor

PLOS ONE

Additional Editor Comments (optional):

The authors addressed all the comments from the reviewers.
---

## [Editor Report · Acceptance letter]

2 Jul 2024

PONE-D-23-39121R2 

PLOS ONE

Dear Dr. Alves, 

I'm pleased to inform you that your manuscript has been deemed suitable for publication in PLOS ONE. Congratulations! Your manuscript is now being handed over to our production team.

Kind regards, 

on behalf of

Prof. Yangsong Zhang 

Academic Editor

PLOS ONE